# Parallelizable Neural Turing Machines

## Abstract

We introduce a parallelizable simplification of Neural Turing Machine (NTM), referred to as P-NTM, which reformulates the core operations of the original architecture as associative operators, enabling the use of efficient parallel scan algorithms. We additionally develop a log-space parallel algorithm for the numerically stable computation of these operations over long sequences. We evaluate the proposed architecture on a synthetic benchmark of algorithmic problems involving state tracking, memorization, and basic arithmetic, solved via autoregressive decoding. We compare P-NTM against a revisited stable implementation of the standard NTM, as well as conventional recurrent and attention-based architectures. Results show that, despite its simplifications, the proposed architecture matches the original in generalization on all evaluated tasks, solving all problems with perfect accuracy, including at unseen sequence lengths. We argue that this is achieved by replacing the recurrent controller with autoregressive control through output tokens. It also exhibits superior training efficiency, with parallel execution being up to an order of magnitude faster than the standard NTM. Ultimately, this work contributes toward the development of efficient neural architectures capable of expressing a broader class of algorithms.

## 1 Introduction

Memory-augmented neural networks offer an alternative to traditional sequence models by integrating explicit memory components. One common class of such networks, inspired by classical models of computation, seeks to combine the expressiveness and structured computation capabilities of these theoretical models with the learnability of neural networks.

These networks typically operate over differentiable memory structures such as stacks (Das et al., 1992; Joulin & Mikolov, 2015), queues (Grefenstette et al., 2015), and tapes (Graves et al., 2014). However, despite their increased expressiveness, these approaches are generally far less efficient than modern language modeling architectures, such as Transformers (Vaswani et al., 2017) and state-space models like Mamba (Gu & Dao, 2024; Dao & Gu, 2024). In particular, their reliance on inherently sequential memory operations makes them difficult to parallelize, significantly limiting their scalability and practical applicability to large-scale training settings.

In response to this, we revisit the Neural Turing Machine (NTM) architecture (Graves et al., 2014). Motivated by recent advances in developing parallelizable recurrent networks (e.g., Feng et al., 2024), we investigate how NTM can be adapted for parallel execution. In this process, we provide three contributions:

- We reexamine the original NTM architecture, reassessing its implementation and initialization hyperparameters, and propose targeted modifications that improve stability and performance (Section 2).

- We introduce the Parallelizable Neural Turing Machine (P-NTM) architecture,[1] which reformulates several operations of the original architecture as associative operators, thereby enabling the use of *parallel scan* algorithms (Blelloch, 1990) for efficient parallelization and effective processing of long input sequences (Section 3.1); and we explain how, despite the simplifications this requires, the

---

[1]An implementation is available at `redacted-for-blind-review`.

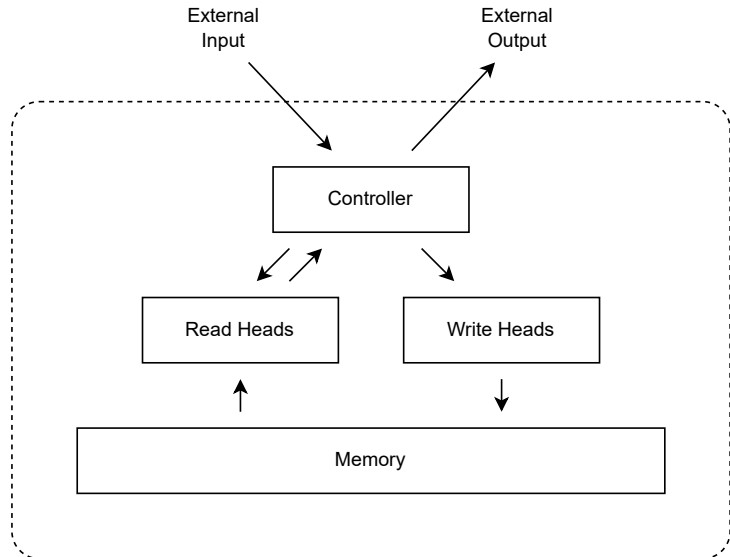

Figure 1: Illustration of the NTM architecture. Adapted from Graves et al. (2014).

new architecture can represent equivalent computations by replacing the recurrent controller with autoregressive control through output tokens (Section 3.4).

- We develop a log-space parallel algorithm for the numerically stable execution of these operations, allowing the model to scale to long sequences without loss of precision (Section 3.2).

We then empirically validate the proposed architecture on a benchmark of synthetic autoregressive algorithmic tasks, evaluating the ability of each architecture to learn and express different kinds of computation, as well as the performance gains enabled by parallelization (Section 4). We show that the proposed model matches the original architecture in generalization performance on the evaluated tasks, with both consistently generalizing to substantially longer, unseen sequence lengths, while providing significant speedup gains enabled by parallelization (Section 5). We further discuss these findings in Section 6. Finally, we outline limitations of the proposed architecture and directions for future work (Section 7) and situate our contributions in relation to prior work (Section 8).

## 2  Neural Turing Machines

NTM (Graves et al., 2014) is a neural network architecture inspired by the classical Turing machine, consisting of a learnable neural controller that interacts with an external memory matrix through fully differentiable read and write operations, which are carried out by a set of read and write heads. Using this mechanism, it can successively interact with memory to perform structured computation over input sequences and represent a variety of algorithmic patterns. An illustration of the architecture is provided in Figure 1.

At each time step, the controller receives the current input along with past information, which includes content retrieved from memory and, in the case of recurrent controllers, a hidden state. Based on the combined input, the controller generates control signals that direct the read and write heads. Some of these signals handle memory addressing, determining where to read from or write to, while others specify the content to be written and erased. After executing the read and write operations, the NTM model generates an output based on its updated internal state.

Below, we present our definition of the NTM architecture and elaborate on its components, with particular emphasis on the modifications introduced in our implementation. This serves both to clearly establish the baseline used in this paper and as a preliminary to our main contribution.

## 2.1 Definition

An NTM can be defined as a sequence-to-sequence mapping that transforms an input sequence of vectors $\boldsymbol{x}_1, \ldots, \boldsymbol{x}_T$, where each $\boldsymbol{x}_t \in \mathbb{R}^d$, into an output sequence $\boldsymbol{y}_1, \ldots, \boldsymbol{y}_T$, where each $\boldsymbol{y}_t \in \mathbb{R}^d$. This mapping is characterized by the following recurrent update equation, applied at each time step $t$:

$$(\boldsymbol{y}_t, \boldsymbol{h}_t, \boldsymbol{M}_t, \boldsymbol{A}_t^r, \boldsymbol{A}_t^w, \boldsymbol{r}_t) = \text{NTM}(\boldsymbol{x}_t, \boldsymbol{h}_{t-1}, \boldsymbol{M}_{t-1}, \boldsymbol{A}_{t-1}^r, \boldsymbol{A}_{t-1}^w, \boldsymbol{r}_{t-1}), \tag{1}$$

where $\boldsymbol{h}_t \in \mathbb{R}^c$ is the internal state of the controller[2] and $\boldsymbol{M}_t \in \mathbb{R}^{m \times n}$ is the memory matrix consisting of $m$ cells[3] of dimension $n$, both at time $t$. Meanwhile, $\boldsymbol{A}_t^r = (\boldsymbol{a}_{t,1}^r, \ldots, \boldsymbol{a}_{t,H}^r)$ and $\boldsymbol{A}_t^w = (\boldsymbol{a}_{t,1}^w, \ldots, \boldsymbol{a}_{t,H}^w)$ respectively denote the addressing weights for each of the $H$ read and write heads[4] at time $t$. Each vector $\boldsymbol{a}_{t,h}^r, \boldsymbol{a}_{t,h}^w \in [0,1]^m$ represents a weighting over the $m$ memory locations, with entries summing to one. These vectors specify how strongly the $h$th read or write head focuses on each memory location. Finally, $\boldsymbol{r}_t \in \mathbb{R}^{Hn}$ is the read vector obtained by aggregating the outputs of the $H$ read heads at time $t$.

The recurrent update (1) that defines the NTM is governed by its control, addressing, memory writing, memory reading, and output mechanisms. Each of these components is formally presented in the following sections. For clarity, we describe the behavior of each read and write head in isolation, noting extensions to the multi-head setting where appropriate.

### 2.1.1 Control

The control mechanism begins by updating the controller based on its previous state, the current input, and the read vector from the previous step. Formally, given a controller function $f_c$, the update is expressed as:

$$\boldsymbol{h}_t = f_c(\boldsymbol{x}_t, \boldsymbol{r}_{t-1}, \boldsymbol{h}_{t-1}), \tag{2}$$

where the initial read vector is set to $\boldsymbol{r}_0 = \boldsymbol{0}$ by default. The controller function $f_c$ may be instantiated as any suitable neural network layer[5]. The resulting controller state $\boldsymbol{h}_t$ is then used to compute the parameters that guide the operations of the NTM at the current step.

For each read or write head, the following set of vectors and scalars is produced to control the addressing mechanism, specifying which memory locations the head will focus on:

$$\boldsymbol{k}_t = \tanh(\boldsymbol{W}_k \boldsymbol{h}_t + \boldsymbol{b}_k), \tag{3}$$

$$\beta_t = \text{softplus}(\boldsymbol{w}_\beta \boldsymbol{h}_t + b_\beta), \tag{4}$$

$$g_t = \sigma(\boldsymbol{w}_g \boldsymbol{h}_t + b_g), \tag{5}$$

$$\boldsymbol{s}_t = \text{softmax}(\boldsymbol{W}_s \boldsymbol{h}_t + \boldsymbol{b}_s), \tag{6}$$

$$\gamma_t = 1 + \text{softplus}(\boldsymbol{w}_\gamma \boldsymbol{h}_t + b_\gamma), \tag{7}$$

where $\boldsymbol{k}_t \in (-1,1)^n$ is the key vector, $\beta_t > 0$ is the key strength, $0 < g_t < 1$ is the interpolation gate, $\boldsymbol{s}_t \in (0,1)^3$ is the shift vector, and $\gamma_t > 1$ is the sharpening factor. Meanwhile, $\boldsymbol{W}_k \in \mathbb{R}^{n \times c}$, $\boldsymbol{W}_s \in \mathbb{R}^{3 \times c}$, $\boldsymbol{w}_\beta, \boldsymbol{w}_g, \boldsymbol{w}_\gamma \in \mathbb{R}^{1 \times c}$ and $\boldsymbol{b}_k \in \mathbb{R}^n$, $b_s \in \mathbb{R}^3$, $b_\beta, b_g, b_\gamma \in \mathbb{R}^1$ are learnable parameters specific to each head. The precise role of each addressing control will be explained in the following sections.

Additionally, for each write head, the following two control vectors are computed to determine how the memory matrix is updated:

$$\boldsymbol{u}_t = \tanh(\boldsymbol{W}_u \boldsymbol{h}_t + \boldsymbol{b}_u), \tag{8}$$

$$\boldsymbol{d}_t = \sigma(\boldsymbol{W}_d \boldsymbol{h}_t + \boldsymbol{b}_d), \tag{9}$$

where $\boldsymbol{u}_t \in (-1,1)^n$ is the add (or update) vector, $\boldsymbol{d}_t \in (0,1)^n$ is the erase (or delete) vector, and $\boldsymbol{W}_u, \boldsymbol{W}_d \in \mathbb{R}^{n \times c}$ and $\boldsymbol{b}_u, \boldsymbol{b}_d \in \mathbb{R}^n$ are learnable parameters specific to each write head. Again, the specific roles of these vectors will be detailed in the subsequent sections.

---

[2] We assume a general class of recurrent models that can be described using a single hidden state vector.

[3] In general, the number of memory cells need not be fixed and can be dynamically adjusted to match the size of the problem at hand.

[4] As a simplifying assumption, we enforce an equal number of read and write heads.

[5] For example, an RNN, an LSTM, or even a feedforward network (if the previous hidden state is disregarded).

### 2.1.2 Addressing

Addressing operates through two primary mechanisms. The first is content-based addressing, which functions similarly to an attention mechanism by focusing on memory cells whose contents closely match a given key vector. The second is location-based addressing, which allows the head to shift to adjacent memory locations, enabling sequential memory access patterns similar to those of a classical Turing machine. These two mechanisms are combined to produce a final vector of addressing weights. The details of this process are outlined below.

**Initialization.** Before processing begins, addressing weights must be initialized to set the initial focus of the heads. While Graves et al. (2014) did not prescribe a specific initialization scheme, subsequent work proposed learning them as a parameter vector (Collier & Beel, 2018). Instead, we initialize all addressing weights to focus solely on the first memory cell: $\boldsymbol{a}_0 = [1 \quad 0 \quad \cdots \quad 0]$.

**Content-based addressing.** Given a similarity function $K : \mathbb{R}^n \times \mathbb{R}^n \to \mathbb{R}$ between vectors, typically instantiated as cosine similarity, the content-based weights $\boldsymbol{a}_t^c$ are computed by measuring the similarity between the current key vector $\boldsymbol{k}_t$ emitted by the controller and each memory cell $\boldsymbol{M}_t[i]$, for $i = 1, \ldots, m$:

$$\boldsymbol{a}_t^c[i] = \frac{\exp(\beta_t \, K(\boldsymbol{k}_t, \boldsymbol{M}_t[i]))}{\sum_{j=1}^m \exp(\beta_t \, K(\boldsymbol{k}_t, \boldsymbol{M}_t[j]))}, \tag{10}$$

where the key strength $\beta_t$ adjusts the sharpness of the weight distribution, controlling how strongly similarity affects the weights.

**Interpolation.** Before proceeding, the newly computed content-based weights are combined with the previous weights using the current interpolation gate $g_t$, yielding the interpolated weights:

$$\boldsymbol{a}_t^g = (1 - g_t)\boldsymbol{a}_{t-1} + g_t \boldsymbol{a}_t^c. \tag{11}$$

This interpolation allows the model to smoothly transition between relying on previous addressing weights and using the new content-based addressing.

**Location-based addressing.** The interpolated weights are then shifted left, right, or kept unchanged according to the shift vector $\boldsymbol{s}_t$ output by the controller at the current time step. This operation yields the shifted weights $\boldsymbol{a}_t^s$, computed via the circular convolution:

$$\boldsymbol{a}_t^s[i] = \sum_{j \in \{-1,0,1\}} \boldsymbol{a}_t^g[i - j] \, \boldsymbol{s}_t[j], \tag{12}$$

where $\boldsymbol{a}_t^g[i - j]$ denotes[6] the interpolated addressing weight at offset $-j$ from position $i$ and $\boldsymbol{s}_t[j]$ is the shift strength associated with offset $j$ (i.e., shift left by 1, no shift, or shift right by 1). This allows the weights to smoothly shift across time steps, wrapping around the memory matrix at its boundaries.

**Sharpening.** Because shifts are not perfectly discrete, their repeated application leads to a blurring effect, where the addressing weights increasingly dissipate. To counteract this, a final sharpening step is applied using the sharpening factor $\gamma_t$, producing the final addressing weights $\boldsymbol{a}_t$. This process can be expressed in two equivalent forms:

$$\boldsymbol{a}_t[i] \overset{(a)}{=} \frac{(\boldsymbol{a}_t^s[i])^{\gamma_t}}{\sum_{j=1}^m (\boldsymbol{a}_t^s[j])^{\gamma_t}} \overset{(b)}{=} \exp\left(\gamma_t \log \boldsymbol{a}_t^s[i] - \log\left(\sum_{j=1}^m \exp(\gamma_t \log \boldsymbol{a}_t^s[j])\right)\right). \tag{13}$$

Crucially, the original formulation (a) of the sharpening step is numerically unstable: moderately large exponents can zero out entries and lead to divide-by-zero errors, leading to `NaN` errors. While prior work has attempted to address this by clipping the outputs of the controller to an arbitrarily chosen range (Collier & Beel, 2018), we instead resolve the issue by adopting formulation (b), implemented with the log-sum-exp trick for numerical stability.

---

[6]The index $i - j$ is treated cyclically over the range 1 to $m$. For example, if $i - j = 0$, the index wraps to $m$; if $i - j = m + 1$, it wraps to 1.

### 2.1.3 Memory Writing

Once the addressing weights for the current step have been computed, the model writes to memory in two steps: first, it selectively erases existing content; then, it adds new content. The erase vector $\boldsymbol{d}_t$ specifies which elements of the memory cells should be cleared, specifically at the locations attended to by the write head. Next, the add vector $\boldsymbol{u}_t$ determines the new content to be added to those same locations. Formally, the update of each memory cell is defined as:

$$\boldsymbol{M}_t[i] = (\mathbf{1} - \boldsymbol{a}_t[i]\,\boldsymbol{d}_t) \odot \boldsymbol{M}_{t-1}[i] + \boldsymbol{a}_t[i]\,\boldsymbol{u}_t, \tag{14}$$

where $\odot$ denotes elementwise vector multiplication.

When multiple write heads are used, all erasures from the different heads are applied first, followed by all corresponding additions. Let $\boldsymbol{d}_{t,1}, \ldots, \boldsymbol{d}_{t,H}$ and $\boldsymbol{u}_{t,1}, \ldots, \boldsymbol{u}_{t,H}$ denote the erase and add vectors from each of the $H$ write heads. Then, the multi-head memory update takes the same form as (14), with a multiplicative erase term $\prod_{h=1}^{H}(\mathbf{1} - \boldsymbol{a}_{t,h}^w[i]\,\boldsymbol{d}_{t,h})$ and an additive term $\sum_{h=1}^{H} \boldsymbol{a}_{t,h}^w[i]\,\boldsymbol{u}_{t,h}$, for each memory location $i = 1, \ldots, m$.

For initialization, we follow the approach of Collier & Beel (2018) where each element of the memory matrix is initialized to a small positive constant by default.

### 2.1.4 Memory Reading

After writing has taken place, the model reads from memory by computing a weighted sum of the updated memory cells using the addressing weights of the read head. This results in the read vector:

$$\boldsymbol{r}_t = \sum_{i=1}^{m} \boldsymbol{a}_t[i]\boldsymbol{M}_t[i], \tag{15}$$

which captures the relevant content from the memory matrix. When multiple read heads are employed, each head independently generates its own read vector by applying its respective addressing weights to memory. These individual read vectors are then concatenated to form the final read vector $\boldsymbol{r}_t$.

### 2.1.5 Output

Finally, the model produces an output vector $\boldsymbol{y}_t$ based on the current controller state and read vector, effectively combining information from the current input as well as both current and past memory. This is computed as:

$$\boldsymbol{y}_t = \boldsymbol{W}_{o,h}\boldsymbol{h}_t + \boldsymbol{W}_{o,r}\boldsymbol{r}_t + \boldsymbol{b}_o, \tag{16}$$

where $\boldsymbol{W}_{o,h} \in \mathbb{R}^{d \times c}, \boldsymbol{W}_{o,r} \in \mathbb{R}^{d \times Hn}, \boldsymbol{b}_o \in \mathbb{R}^d$ are learnable parameters.

## 3 Parallelizable Neural Turing Machines

P-NTM is a simplified and parallelizable variant of the standard NTM. It retains the central concept of a recurrent model with multiple read and write heads operating over a memory matrix, but introduces key simplifications primarily aimed at enabling efficient parallel computation.

Most notably, it forgoes content-based addressing and eliminates temporal dependencies in the control mechanism. That is, head controls at each time step depend solely on the current input and are unaffected by past memory or control signals. As a result, this design enables internal states to be computed in parallel across time steps using scan operations.

In the following sections, we present the P-NTM architecture in detail. We begin by introducing its recurrent formulation (Section 3.1) and then explain how it enables parallel execution (Section 3.2). Next, we present a stabilization mechanism aimed at improving inference over long sequences (Section 3.3). Finally, we discuss how P-NTM can represent computation and contrast it with the standard NTM (Section 3.4).

### 3.1 Recurrent Formulation

A P-NTM can be formulated analogously to the standard NTM described in Section 2.1, but with a simplified update equation:

$$(\boldsymbol{y}_t, \boldsymbol{M}_t, \boldsymbol{A}_t^r, \boldsymbol{A}_t^w) = \text{P-NTM}(\boldsymbol{x}_t, \boldsymbol{M}_{t-1}, \boldsymbol{A}_{t-1}^r, \boldsymbol{A}_{t-1}^w), \tag{17}$$

where dependencies on the recurrent controller state and the previous read vector are removed, so that the model maintains only the memory matrix and two sets of addressing weights for the read and write heads.

Specifically, at each step, the model computes head controls based on the current input. These controls determine how the read and write heads shift their focus over the memory matrix as well as what content should be written to memory. Writes are non-interfering, with each write head targeting a distinct portion of each memory cell. A learnable mixing mechanism allows each read head to access content from any write head. The final output is computed by projecting the aggregated readings from the read heads.

We now describe the components of this recurrent update in detail. As before, we consider a single pair of read and write heads in isolation, noting extensions to the multi-head scenario where applicable. The full multi-head implementation is provided in Appendix A (Algorithm 3).

#### 3.1.1 Control

For each pair of read and write heads, the control mechanism generates three components based on the inputs: a read shift vector, a write shift vector, and a memory update vector. These are respectively defined as:

$$\boldsymbol{s}_t^r = \text{softmax}(\boldsymbol{W}_r \boldsymbol{x}_t), \tag{18}$$
$$\boldsymbol{s}_t^w = \text{softmax}(\boldsymbol{W}_w \boldsymbol{x}_t), \tag{19}$$
$$\boldsymbol{u}_t = \boldsymbol{W}_u \boldsymbol{x}_t, \tag{20}$$

where $\boldsymbol{W}_r, \boldsymbol{W}_w \in \mathbb{R}^{3 \times d}$ and $\boldsymbol{W}_u \in \mathbb{R}^{n \times d}$ are learnable parameters specific to each pair of heads. The shift vectors $\boldsymbol{s}_t^r, \boldsymbol{s}_t^w \in (0,1)^3$ represent the strength of different head shift operations (i.e., left, stay, right), while the update vector $\boldsymbol{u}_t \in \mathbb{R}^n$ specifies the new content to be written to memory in the current step.

#### 3.1.2 Addressing

The addressing mechanism follows the location-based strategy of the standard NTM. By design, the initial weights are fully concentrated on the first memory cell: $\boldsymbol{a}_0 = [1 \quad 0 \quad \cdots \quad 0]$. At each subsequent time step, the addressing weights of each head are updated via circular convolution using the previous weights and the current shift vector. For each memory location $i = 1, \ldots, m$:

$$\boldsymbol{a}_t[i] = \sum_{j \in \{-1,0,1\}} \boldsymbol{a}_{t-1}[i-j] \, \boldsymbol{s}_t[j], \tag{21}$$

where $\boldsymbol{a}_{t-1}[i-j]$ is indexed cyclically and $\boldsymbol{s}_t[j]$ denotes the shift strength for offset $j$, as in (12).

A subtle but important detail is that both reading and writing operations in P-NTM rely on the addressing weights $\boldsymbol{a}_{t-1}$ from the previous time step, rather than the current weights $\boldsymbol{a}_t$. As a result, the addressing weights are typically updated as the final operation in each recurrent computation step. This design choice reflects the behavior of classical Turing machines, where reading and writing occur at the location left by the previous transition.

#### 3.1.3 Memory Writing

The memory matrix is first initialized to zeros: $\boldsymbol{M}_0 = \boldsymbol{0}$. At each time step, the contents of each memory cell are updated using the previous write addressing weights and the current update vector:

$$\boldsymbol{M}_t[i] = (1 - \boldsymbol{a}_{t-1}[i])\boldsymbol{M}_{t-1}[i] + \boldsymbol{a}_{t-1}[i] \, g(\boldsymbol{u}_t), \tag{22}$$

where $g$ is a nonlinearity introduced by Feng et al. (2024), defined elementwise as:

$$g(x) = \begin{cases} x + 0.5, & \text{if } x \geq 0 \\ \sigma(x), & \text{otherwise.} \end{cases} \tag{23}$$

This nonlinearity ensures that the memory contents remain differentiable, non-negative, and unbounded above. These characteristics, in turn, enable an effective log-space representation of memory, a property that will become important later.

After the update, each memory cell is mixed through a linear projection:

$$\tilde{\boldsymbol{M}}_t[i] = \boldsymbol{W}_m \boldsymbol{M}_t[i], \tag{24}$$

where $\boldsymbol{W}_m \in \mathbb{R}^{n \times n}$ is a learnable matrix.

In the multi-head setting, each memory cell is partitioned among multiple write heads. For instance, with a 12-dimensional cell and 3 write heads, dimensions 1–4 go to head 1, 5–8 to head 2, and 9–12 to head 3. This arrangement ensures that each head writes to a distinct portion of the memory cell, avoiding interference between them. Yet, because of the projection in (24), information written by a single head is distributed across the entire memory cell, allowing interaction between the contributions of different heads.

### 3.1.4 Memory Reading

After writing, the model retrieves information from memory by computing a weighted sum over the mixed memory cells using the addressing weights of the read head from the previous step, producing the read vector:

$$\boldsymbol{r}_t = \sum_{i=1}^{m} \boldsymbol{a}_{t-1}[i]\, \tilde{\boldsymbol{M}}_t[i]. \tag{25}$$

With multiple read heads, each head performs a separate read using its own addressing weights, and their read vectors are concatenated to form the final vector $\boldsymbol{r}_t$.

### 3.1.5 Output

The final output at each time step is computed by linearly transforming the read vector:

$$\boldsymbol{y}_t = \boldsymbol{W}_o \boldsymbol{r}_t, \tag{26}$$

where $\boldsymbol{W}_o \in \mathbb{R}^{d \times Hn}$ is a learnable output projection matrix. In the multi-head setting, the concatenated read vectors from all heads are jointly transformed by this matrix, enabling the model to integrate information from multiple heads into the final output.

### 3.2 Parallel Implementation

In the parallel execution of P-NTM, computation proceeds in a few well-defined stages. First, all input vectors are processed simultaneously to produce the shift and update control vectors. The shift controls are then converted into read and write addressing vectors using a parallel scan operation. Next, given the write addresses and update vectors, another scan generates the memory states for all sequence positions. Finally, the read vectors are computed in parallel from the memory states and read addresses, followed by the construction of the output sequence.

Notably, parallelism in P-NTM arises in two main forms. Position-wise parallelism occurs when operations at each sequence position, such as computing the control vectors, are independent and can be executed simultaneously. Scan parallelism handles the recurrences in addressing and memory updates by reformulating them as prefix computations, allowing cumulative operations to be efficiently calculated across the sequence. The overall structure of this parallel computation strategy is illustrated in Figure 2. The work, depth, and space complexity of P-NTM execution are analyzed in Appendix A.

Below, we introduce the scan-based primitive used in P-NTM and explain how it enables parallel computation of addressing weights and memory states across time steps.

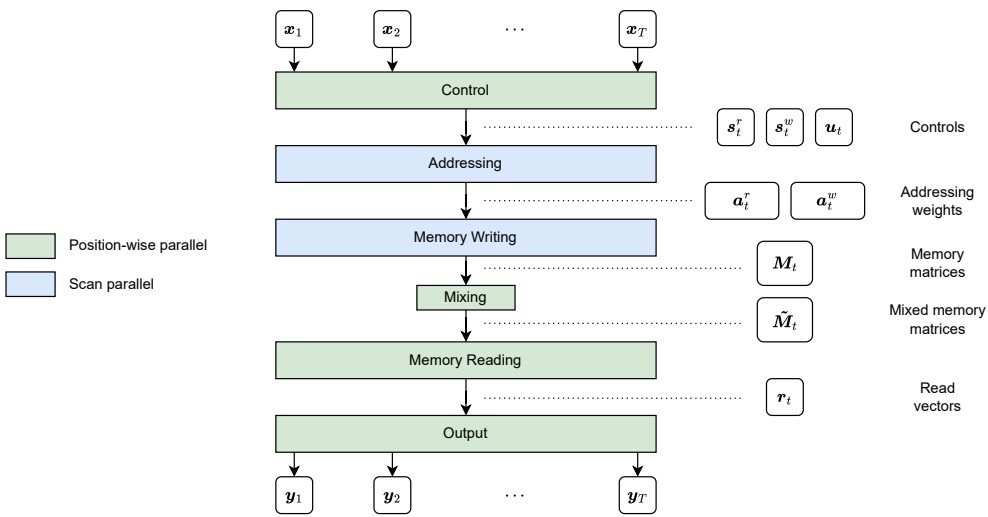

Figure 2: Illustration of the parallel computation of P-NTM.

### 3.2.1 Parallel Scan

The parallel scan algorithm (Blelloch, 1990) enables the efficient computation of all prefix values of a sequence, obtained by repeatedly applying an associative binary operator $\oplus$ such as addition or multiplication. Given a sequence of inputs $u_1, u_2, \ldots, u_T$, the scan operation produces the prefix sequence:

$$u_1, (u_1 \oplus u_2), \ldots, (u_1 \oplus u_2 \oplus \cdots \oplus u_T). \tag{27}$$

Because of the associativity of the binary operator, the computation can be structured as a tree, allowing it to be distributed across multiple processors. Relative to sequence length, this parallelization achieves linear speedup with respect to the number of processors, introducing only logarithmic overhead.

The scan operation serves as a basic primitive for a broad class of computations, including cumulative sums, cumulative products, and, generally, first-order recurrences of the form:

$$v_t = \alpha_t v_{t-1} + \beta_t, \tag{28}$$

where $\alpha_t$ and $\beta_t$ are real-valued scalars. Given the initial value $v_0$ and the sequences $\alpha_1, \ldots, \alpha_T$ and $\beta_1, \ldots, \beta_T$, the scan algorithm can compute the entire sequence $v_1, \ldots, v_T$ in parallel. As we will see, these computations form the basis of both the addressing and memory update operations in P-NTM, which in turn enables them to be executed in parallel.

### 3.2.2 Shift-based Addressing

As previously mentioned, updating addressing weights using a shift vector from one time step to the next corresponds to a circular convolution operation. This perspective enables us to leverage the mathematical properties of convolution to efficiently compute sequences of shifts, allowing for parallel construction of addressing vectors across time steps.

According to the convolution theorem (Oppenheim et al., 1999), the circular convolution between two sequences (or vectors) $\boldsymbol{x}$ and $\boldsymbol{y}$ can be computed through an elementwise product in the frequency domain using the fast Fourier transform (FFT) and its inverse (IFFT):

$$\boldsymbol{x} * \boldsymbol{y} = \text{IFFT}(\text{FFT}(\boldsymbol{x}) \odot \text{FFT}(\boldsymbol{y})). \tag{29}$$

We use this fact to compute addressing in parallel. Let $\boldsymbol{k}_t \in \mathbb{R}^m$ be the kernel corresponding to the shift $\boldsymbol{s}_t$, obtained by padding the shift vector to the memory size $m$ and circularly shifting its elements to the left.

---

**Algorithm 1** $\boldsymbol{a}_0, \ldots, \boldsymbol{a}_{T-1} \leftarrow \texttt{ConvShift}(\boldsymbol{s}_1, \ldots, \boldsymbol{s}_{T-1})$

---

   **Input:** $\boldsymbol{s}_1, \ldots, \boldsymbol{s}_{T-1}$, sequence of shift vectors, where each $\boldsymbol{s}_t \in \mathbb{R}^3$.
   **Output:** $\boldsymbol{a}_0, \ldots, \boldsymbol{a}_{T-1}$, sequence of addressing vectors, where each $\boldsymbol{a}_t \in \mathbb{R}^m$.
 1: **parallel for** $t \leftarrow 0$ **to** $T - 1$ **do**
 2:     $\boldsymbol{k}_t \leftarrow \text{roll}_{-1}(\text{pad}_m(\boldsymbol{s}_t))$ **if** $t > 0$ **else** $[1, 0, \ldots, 0]$[7]
 3:     $\boldsymbol{\psi}_t \leftarrow \widetilde{\log}(\text{FFT}(\boldsymbol{k}_t))$
 4: **end parallel for**
 5: $\tilde{\boldsymbol{\phi}}_0, \ldots, \tilde{\boldsymbol{\phi}}_{T-1} \leftarrow \text{cumsum}(\boldsymbol{\psi}_0, \ldots, \boldsymbol{\psi}_{T-1})$
 6: **parallel for** $t \leftarrow 0$ **to** $T - 1$ **do**
 7:     $\boldsymbol{a}_t \leftarrow \text{IFFT}(\exp(\tilde{\boldsymbol{\phi}}_t))$
 8:     $\boldsymbol{a}_t \leftarrow \text{clamp}_{[0,1]}(\boldsymbol{a}_t)$
 9: **end parallel for**
10: **return** $\boldsymbol{a}_0, \ldots, \boldsymbol{a}_{T-1}$

---

Then, the addressing vector at time $t$ can be computed as:

$$\boldsymbol{a}_t = \text{IFFT}(\boldsymbol{\phi}_t), \tag{30}$$

$$\boldsymbol{\phi}_t = \text{FFT}(\boldsymbol{a}_0) \odot \text{FFT}(\boldsymbol{k}_1) \odot \cdots \odot \text{FFT}(\boldsymbol{k}_{t-1}). \tag{31}$$

This formulation enables the computation of addressing weights at each time step by applying an inverse Fourier transform to the cumulative elementwise product of the transformed kernels. Importantly, as discussed in the previous section, this product admits a parallel implementation.

However, this method can suffer from numerical underflow due to the repeated multiplication of small-magnitude values. Since both addressing weights and shift vectors have entries less than or equal to one, their corresponding frequency-domain representations may quickly decay when multiplied over many time steps, leading to vanishingly small values.

To address this, we revisit the convolution computation. Rather than performing the operation directly, we compute the convolution in log-space by replacing elementwise multiplications with sums of approximate complex logarithms. The resulting approximation is given by:

$$\boldsymbol{x} * \boldsymbol{y} \approx \text{IFFT}(\exp(\widetilde{\log}(\text{FFT}(\boldsymbol{x})) + \widetilde{\log}(\text{FFT}(\boldsymbol{y})))), \tag{32}$$

where the approximate logarithm is applied elementwise as:

$$\widetilde{\log}(x) = \begin{cases} \log(x + \varepsilon \cdot \text{sign}(x)), & \text{if } x \neq 0 \\ \log(\varepsilon), & \text{otherwise.} \end{cases} \tag{33}$$

Here, $\varepsilon$ is a small positive constant and the complex sign function denotes the unit vector in the direction of $x$. This approximation avoids the undefined behavior of the complex logarithm at zero and mitigates the instability caused by large gradients near zero.

Ultimately, this approach enables a stable approximation of each addressing vector through a cumulative sum that can be parallelized across the sequence length, such that:

$$\boldsymbol{a}_t \approx \text{IFFT}(\exp(\tilde{\boldsymbol{\phi}}_t)), \tag{34}$$

$$\tilde{\boldsymbol{\phi}}_t = \widetilde{\log}(\text{FFT}(\boldsymbol{a}_0)) + \widetilde{\log}(\text{FFT}(\boldsymbol{k}_1)) + \cdots + \widetilde{\log}(\text{FFT}(\boldsymbol{k}_{t-1})). \tag{35}$$

The complete implementation is detailed in Algorithm 1, which includes an additional clamping step to account for approximation errors in the addressing weights, keeping them within the valid range of 0 to 1.

---

[7]Here, $\text{pad}_m(\cdot)$ extends the 3-element shift vector to length $m$ by appending zeros, and $\text{roll}_{-1}(\cdot)$ circularly shifts the vector by $-1$ (i.e., one position to the left). This ensures that the FFT-based circular convolution applies the shift kernel according to the intended left-stay-right weighting.

---

**Algorithm 2** $M_1, \ldots, M_T \leftarrow \texttt{MemoryWrite}(a_0, \ldots, a_{T-1}, u_1, \ldots, u_T)$

---

**Input:** $a_0, \ldots, a_{T-1}$, sequence of addressing vectors, where each $a_t \in \mathbb{R}^m$;
      $u_1, \ldots, u_T$, sequence of update vectors, where each $u_t \in \mathbb{R}^n$.
**Output:** $M_1, \ldots, M_T$, sequence of memory matrices, where each $M_t \in \mathbb{R}^{m \times n}$.

1: **parallel for** $t \leftarrow 1$ **to** $T$ **do**                                ▷ Compute scan coefficients.
2:      $\bar{a}_t \leftarrow \text{clamp}_{[\varepsilon, 1-\varepsilon]}(a_{t-1})$
3:      $\alpha_t \leftarrow \log(1 - \bar{a}_t)$
4:      $B_t \leftarrow \log \bar{a}_t + \log g(u_t)^\top$
5: **end parallel for**
6: $\alpha_1^\star, \ldots, \alpha_T^\star \leftarrow \text{cumsum}(\alpha_1, \ldots, \alpha_T)$                          ▷ Apply scan.
7: $H_1, \ldots, H_T \leftarrow \text{cumlogsumexp}(B_1 - \alpha_1^\star, \ldots, B_T - \alpha_T^\star)$
8: $M_1, \ldots, M_T \leftarrow \exp(\alpha_1^\star + H_1), \ldots, \exp(\alpha_T^\star + H_T)$
9: **return** $M_1, \ldots, M_T$

---

### 3.2.3 Memory Writes

From the memory update rule in (22), each scalar entry of the memory matrix evolves according to a first-order recurrence, as defined in (28), with $v_0 = 0$, $\alpha_t = 1 - a_{t-1}[i]$, and $\beta_t = a_{t-1}[i] \, g(u_t[j])$, for each memory cell $i$ and component $j$. Because all $\alpha_t$ and $\beta_t$ are known in advance (since all addressing and update vectors are known at this point), the parallel scan algorithm can be applied to compute memory updates efficiently over the entire sequence.

To ensure numerical stability and maintain precision over long sequences, we follow the method used by Feng et al. (2024), which builds on the log-space parallel scan algorithm of Heinsen (2023). This approach computes the recurrence using cumulative sums in log-space, allowing for efficient and stable evaluation. In our case, the log-space representation is approximate, as exact zeros in the addressing weights cannot be expressed. Nonetheless, it remains an effective implementation.

The complete parallel memory writing procedure is described in Algorithm 2. Write addressing weights and their complements are broadcast across memory cells to enable elementwise updates. After parallel updates are applied, each memory cell is linearly projected as described in (24).

### 3.3 Stable Inference

As discussed earlier, because shifts are generated by a softmax and thus not perfectly discrete, their repeated application leads to a blurring effect, causing the addressing weights to progressively dissipate. While the standard NTM addresses this issue using a learnable sharpening mechanism, we instead apply a separate stabilization step during inference.

Specifically, any shift strength below a threshold hyperparameter $0 \leq \tau < 1$ is set to zero. The remaining shift strengths, which exceed $\tau$, are then recomputed by applying the softmax only over their corresponding offsets. The rationale is that we may interpret extremely small shift weight values as a result of the mathematical inability of the softmax function to produce exact zeros, combined with a difficulty of the optimizer in converging toward such distributions. Therefore, sufficiently small values of $\tau$ may allow discrete shift patterns to be effectively represented by suppressing shift directions with negligible weights.

Finally, we note that inference is performed recurrently in linear space, following the equations presented in Section 3.1, rather than the log-space parallel implementation described in Section 3.2. This is so that discrete behavior can be properly represented.

### 3.4 Computational Mechanism

Standard NTMs perform computation via a nonlinear recurrent controller that maintains a hidden state across time steps, serving as the finite-state control mechanism analogous to that of a classical Turing machine. At each step, the controller combines the current input, information retrieved from memory,

Table 1: Algorithmic tasks with input and output examples. Final answers are underlined. Detailed descriptions of each task are provided in Appendix B.

| Task | Example Input | Example Output |
|---|---|---|
| Parity Check (PC) | *aaabba* | 01000$\underline{1}$ |
| Cycle Navigation (CN) | *siidis* | 01212$\underline{2}$ |
| Reverse String (RS) | *aabba* | $\underline{abbaa}$ |
| Duplicate String (DS) | *aabba* | $\underline{aabbaaabba}$ |
| Modular Arithmetic (MA) | $1 + 2 - 4$ | $+10+21-43\underline{4}$ |
| Binary Addition (BA) | $01101 + 101$ | $\underline{11011}$ |

and its previous hidden state to generate control signals governing read and write operations. Continuous nonlinear recurrent networks of the form used in NTMs, as specified in (2), are known to be capable of representing finite automata (Siegelmann, 1996; Weiss et al., 2018). This capability enables the controller to realize the finite-state component of a Turing machine, while the memory matrix provides an analog of the tape.

P-NTM, by contrast, departs from this design by eliminating the explicit controller state. All control decisions are generated solely from the current input, with no direct dependence on memory contents or recurrent hidden states. This architectural choice prevents the model from tracking the finite-state component of computation via an internal state register, as in standard NTMs, and therefore requires computation to be realized through an alternative mechanism.

Instead, P-NTM achieves structured computation through autoregression. The model consumes its own outputs as subsequent inputs while continuing to read from and write to the memory matrix. In this setting, the current input token (typically the most recent output) acts as an implicit state variable that conditions control at each step. State information is therefore encoded in the output sequence itself, rather than in a persistent hidden vector, effectively distributing state management across time and eliminating the need for a nonlinear recurrent controller. The following sections present empirical experiments and results that illustrate the computational capabilities of this autoregressive formulation.

## 4 Experiments

We evaluate the proposed P-NTM architecture in two ways. First, we empirically assess its ability to learn and represent various algorithmic behaviors in an autoregressive setting (i.e., via *chain of thought*), despite the simplifications introduced relative to the original design. Second, we verify the computational efficiency gains enabled by its parallelism compared to the original architecture. The general experimental setup for each evaluation is described in the following sections, with further details being provided in Appendix B.

### 4.1 Algorithm Learning and Representation

The experimental design for assessing learning and representation of algorithms follows that of Delétang et al. (2023). It consists of training models with different architectures on various algorithmic tasks (e.g., evaluating an arithmetic expression) and determining their ability to generalize to larger, previously unseen instances of those tasks. A model is considered to have successfully learned the underlying algorithm if it can consistently solve instances larger than those seen during training. The rationale for these tasks is to simulate, in a controlled and simplified context, the reasoning and computational challenges typically found in real-world problems, which are often expressed in natural language and addressed by large language models.

**Tasks.** Each algorithmic task consists of taking an input sequence of tokens, which represents a particular problem instance, followed by a special separator token, and producing an output sequence of tokens representing its solution. The output is generated step by step, including all intermediate tokens required to reach the final answer. The tasks incorporated into our experiments are a subset of those from Delétang et al.

(2023), further modified to include explicit intermediate steps for every problem. They are intended to probe computational capabilities such as state-tracking, memorization, and basic arithmetic. Table 1 summarizes the tasks and provides representative input and output examples.

**Architectures.** Several neural network architectures are evaluated in the experiment, including not only the neural Turing machines central to this work but also standard sequence models. This broader comparison provides clearer context for interpreting the results. Specifically, both standard recurrent and parallelizable linear recurrent architectures are included as baselines, using LSTM (Hochreiter & Schmidhuber, 1997) and minimal gated recurrent unit (minGRU) (Feng et al., 2024) networks, respectively. Autoregressive Transformers (Vaswani et al., 2017) are also assessed under different position encoding schemes, including NoPE (no positional encoding), ALiBi (Press et al., 2022), RoPE (Su et al., 2024), and FIRE (Li et al., 2024a). The evaluation further includes a standard NTM with an LSTM controller and a P-NTM preceded by a minGRU layer. In this configuration, the minGRU serves as a parallelizable recurrent aggregator of past information, aiding P-NTM's control mechanism, which would otherwise only consider the current token. All models share a common structure consisting of an initial token-embedding layer and a final softmax layer for computing output token probabilities. They are also designed to have comparable parameter counts.

**Training.** Models are trained on randomly sampled batches of problems for up to 500,000 iterations. At each iteration, an input length $\ell \in [1, 40]$ is chosen uniformly at random, and 128 examples of that length are drawn. Each training example consists of the input and output sequences concatenated with the separator token. Training uses supervised learning with cross-entropy loss on the output tokens, conditioning each prediction on the previous ground-truth token. The Adam optimizer (Kingma & Ba, 2017) is used, with the learning rate fixed at $5 \times 10^{-4}$ for the entire duration of training. Early stopping is triggered if the infinity norm of the parameter gradient remains below $10^{-8}$ for 500 consecutive iterations. To assess sensitivity to initialization, ten models of each architecture are trained per task, each with a different random seed.

**Evaluation.** Each trained model is evaluated using greedy decoding to generate outputs from the inputs. While more sophisticated decoding strategies could yield better results, we adopt greedy decoding as a simple baseline. Generalization performance is measured using *exact match accuracy* on problems with input lengths $\ell \in [41, 120]$, corresponding to inputs up to three times longer than those observed during training. For each value of $\ell$, accuracy is estimated using 128 randomly sampled inputs and is computed as the proportion of generated output sequences that exactly match the target sequence. For models with external memory (NTM and P-NTM), the memory size is increased relative to the training configuration to accommodate the larger problem sizes used during evaluation.

## 4.2 Computational Efficiency

To evaluate computational efficiency, the experiment measures how forward-pass execution time scales with input length for both a standard NTM and the proposed P-NTM. For P-NTM, we measure both sequential and parallel execution times to isolate the impact of parallelization from other architectural simplifications. The goal is to quantify the speed advantage offered by parallel computation, particularly for long sequences.

The NTM model in this experiment uses an LSTM controller, while the P-NTM model employs a preceding minGRU layer, following the architectures described in the previous section. Both models use a single read–write head pair, a hidden dimension of 128, and a fixed memory size of 512. To match the parameter count of the LSTM controller, the minGRU layer in the P-NTM model uses a state expansion factor of 3.

Input lengths ranging from $2^3 = 8$ to $2^{16} = 65{,}536$ are tested, increasing in powers of two. For each length, a batch of 8 synthetic sequences of 128-dimensional vectors is generated, and execution time is measured over the batch. Then, 10 timed runs are performed to obtain the runtime measurements, preceded by 3 warm-up runs to account for compilation and caching effects. Finally, the mean speedup of P-NTM relative to NTM is reported for both sequential and parallel execution, calculated as the ratio of their respective mean runtimes. All measurements are obtained on an NVIDIA A100 system, providing a high-throughput environment for runtime comparisons.

Table 2: Maximum exact match accuracy scores across runs for each architecture and task on problems of unseen length ($\ell = 41$ to $\ell = 120$). Perfect scores are highlighted in bold.

| Architecture | PC | CN | RS | DS | MA | BA |
|---|---|---|---|---|---|---|
| LSTM | 0.09 | 0.04 | 0.14 | 0.09 | 0.06 | 0.12 |
| minGRU | 0.07 | 0.04 | 0.09 | 0.09 | 0.08 | 0.10 |
| Transformer (best) | 0.27 | 0.56 | 0.21 | 0.65 | 0.42 | 0.22 |
| NTM | **1.00** | **1.00** | **1.00** | **1.00** | **1.00** | **1.00** |
| P-NTM | **1.00** | **1.00** | **1.00** | **1.00** | **1.00** | **1.00** |

## 5 Results

### 5.1 Length Generalization

Table 2 reports the maximum exact match accuracy achieved across runs for each architecture[8] on each task when evaluated on sequences of unseen length. This metric indicates whether any training run found a parameter configuration capable of extrapolating beyond the training data and solving instances substantially larger than those observed during training.

The results show that, like the standard NTM, P-NTM successfully learns all evaluated computational tasks, with both architectures achieving perfect accuracy across all settings. In contrast, the recurrent baselines (LSTM and minGRU) perform poorly, with accuracy at most 0.14 across all tasks. Transformer models generalize better than recurrent networks but still fall short of the Turing-inspired architectures, achieving at most 0.65 accuracy.

### 5.2 Sensitivity to Initialization

Figure 3 shows, for each task, the distribution of exact match accuracies obtained under different initialization seeds. This highlights variability in learning outcomes and reveals how reliably each architecture converges to solutions that generalize to unseen sequence lengths.

Both NTM and P-NTM achieve perfect generalization in a fraction of runs (35% and 55%, respectively), but not consistently. When performance does degrade, the two architectures fail differently. P-NTM exhibits a lower mean accuracy (0.93 vs. 0.97) and a larger standard deviation (0.14 vs. 0.11) across runs (Appendix C, Table 6), with accuracy dropping abruptly beyond a certain length rather than degrading gradually (Appendix C, Figure 7).

By contrast, recurrent architectures largely fail to generalize to longer lengths, while Transformers exhibit mixed behavior: some runs achieve partial generalization, but none match the consistency of the memory-augmented models.

### 5.3 Speedup

Figure 4 shows the speedup achieved by both parallel and sequential P-NTM execution relative to the standard NTM as input length increases. Sequential P-NTM execution attains a speed comparable to that of the standard NTM across all input lengths. In contrast, parallel P-NTM consistently outperforms the baseline, with execution running $3.6\times$ to $18.5\times$ faster than the standard NTM, and speedup increasing with input length. The timings for each architecture and input length are provided in Appendix C (Table 7).

## 6 Discussion

The central question surrounding P-NTM is whether the recurrent controller of the standard NTM can be removed without sacrificing the ability to learn algorithmic tasks. On the evaluated benchmarks, P-NTM

---

[8]For Transformers, only the highest-scoring positional encoding variant is shown. Full results are provided in Appendix C.

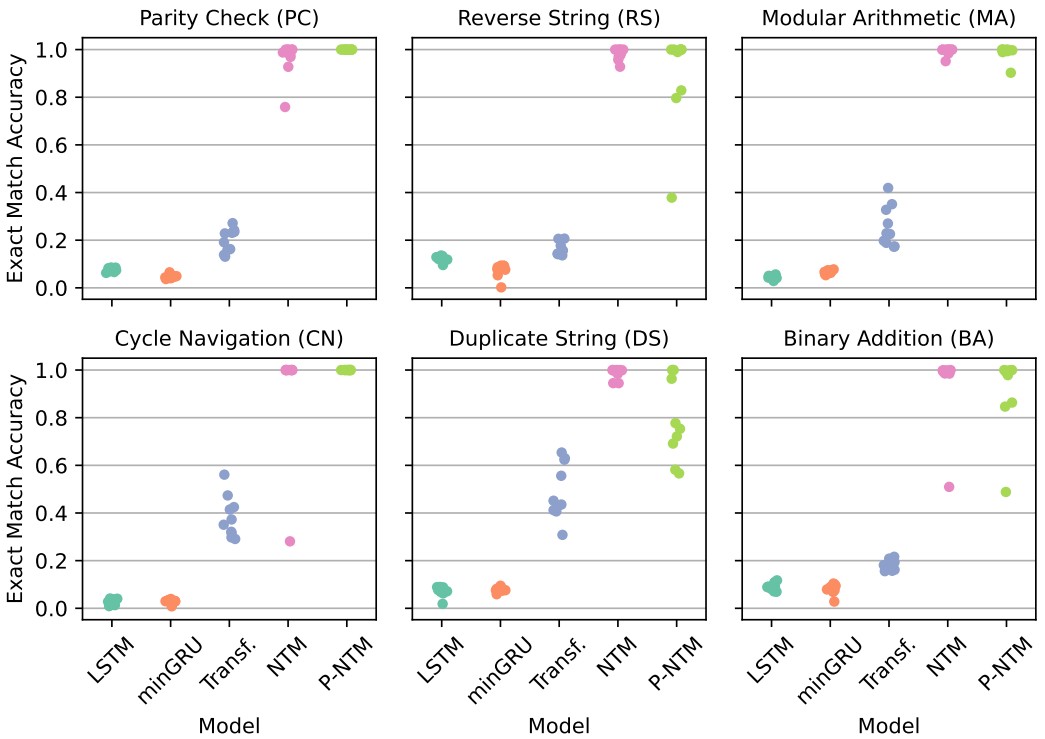

Figure 3: Exact match accuracy scores across runs for different architectures and tasks on problems of unseen length ($\ell = 41$ to $\ell = 120$). For Transformers, the positional encoding variant with the highest maximum exact match accuracy per task is shown (see Table 5).

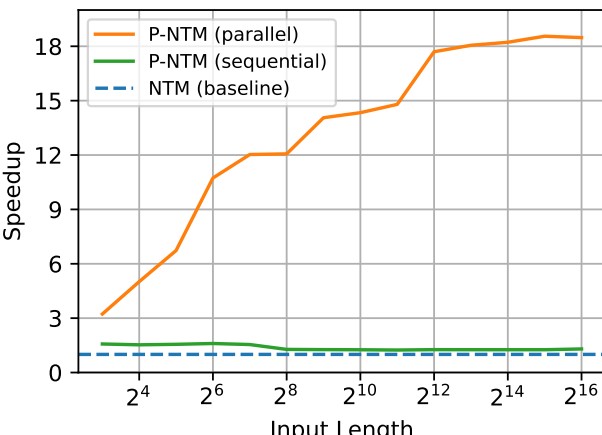

Figure 4: Speedup of P-NTM sequential and parallel execution relative to the standard NTM execution across input batches of varying sequence lengths.

matches the peak accuracy of the standard NTM on every task, demonstrating that explicit controller state is not required for these tasks when external memory is available and control is mediated autoregressively through output tokens. Gradient-based optimization alone is sufficient to discover effective memory access strategies under this design.

Comparing across architecture classes, the memory-augmented models (NTM and P-NTM) consistently outperform both simple recurrent baselines and Transformers on length generalization. Transformers do improve over recurrent baselines, but fail to reliably generalize to longer sequences, consistent with prior findings on algorithmic tasks (Delétang et al., 2023; Zheng et al., 2024). This suggests that external memory plays an important role in supporting the kind of systematic generalization these tasks require.

While P-NTM matches the standard NTM in peak accuracy, the two architectures differ in how they fail. When generalization degrades, P-NTM exhibits abrupt drops in accuracy rather than the gradual decline observed in the standard NTM (Appendix C, Figure 7). One possible explanation is that the threshold-based stabilization does not fully prevent small addressing errors from compounding over long sequences.

On the efficiency side, the speedups from parallelism grow with sequence length and account for the majority of the gains over the standard NTM, beyond what the architectural simplifications alone provide. This comes at the cost of memory: standard NTM inference uses constant space per step, whereas parallel P-NTM execution via scan requires space that grows linearly with sequence length (Appendix A, Table 3), so full parallelization is bounded by available memory and very long sequences may require chunked execution.

## 7 Limitations

A central tradeoff of P-NTM is its reliance on autoregressive interaction with output tokens to represent and update computational state. This design performs well during training when problems are paired with sufficiently detailed intermediate reasoning steps, enabling the model to learn these computations in parallel. But when such intermediate outputs are missing, sparse, or noisy, the model may struggle to reliably learn and track discrete states (Merrill et al., 2024). In contrast, standard NTMs maintain state internally within a recurrent controller and can, at least in principle, learn state-tracking behavior even in the absence of intermediate supervision, such as when training data consists of blank or uninformative tokens.

One potential direction for mitigating this limitation is to explore augmenting P-NTM with more expressive linear recurrent architectures. Recent work on the DeltaNet family of models shows that such architectures can represent and learn regular languages while remaining parallelizable (Grazzi et al., 2025; Yang et al., 2024; Siems et al., 2025). Integrating similar components into a layer preceding P-NTM or into its recurrence could allow the model to retain its parallelism benefits while improving its ability to track state implicitly. This, in turn, may reduce reliance on explicit intermediate reasoning steps in the training data.

A related open question concerns the formal computational model of P-NTM. While standard NTMs have a relatively direct correspondence to classical Turing machines, the model of computation realized by P-NTM under autoregressive operation has not been formally characterized. As such, it is not yet exactly known what intermediate reasoning sequences are sufficient or necessary for a given computational problem.

At the same time, the parallelism advantages of P-NTM do not carry over to settings where computation is inherently sequential. In autoregressive generation regimes, including some online reinforcement learning scenarios, per-step computation cost is the dominant bottleneck, and scalability is instead limited by the size of the memory matrix and the cumulative cost of generation steps. More efficient memory representations and addressing mechanisms could help mitigate these costs.

## 8 Related Work

**Turing-inspired neural networks.** Our work is related to several extensions of the original NTM architecture. Differentiable Neural Computer (DNC) (Graves et al., 2016) introduces additional memory management mechanisms, including a temporal link matrix and memory usage tracking, which together enable more flexible location-based access patterns and mitigate write interference. Dynamic Neural Turing Machine (D-NTM) (Gülçehre et al., 2017), on the other hand, aims to improve the flexibility of memory access patterns by introducing fixed, learnable address vectors and a recency-based access strategy, with both continuous and discrete variants. Meanwhile, our work is orthogonal to these extensions and instead focuses on parallelization, rather than on increasing memory flexibility or stability. More similar to our work is Baby-NTM (Suzgun et al., 2019), which simplifies the NTM by removing content-based addressing

and retaining only location-based addressing. Unlike Baby-NTM, however, our method introduces a targeted simplification of the NTM that enables parallel computation. More broadly, Neural GPUs (Kaiser & Sutskever, 2016) pursue a goal similar to ours and seek to address the sequential bottleneck of NTMs. They do so by removing explicit memory operations and instead iteratively applying convolutional layers over the entire input sequence to produce the output. While this design enables fully parallel computation, it does not naturally support recurrent execution for autoregressive generation. In contrast, our approach preserves recurrent execution while still enabling parallel computation. Finally, nnTM (Stogin et al., 2024) is a Turing-complete architecture based on higher-order connections and stack-based memory, with provable inference-time stability guarantees. Instead, we retain an explicit memory matrix with moving heads, as in NTMs, while adopting a parallelizable architecture. Inference stability is encouraged through discrete autoregressive outputs, as well as clipping and renormalization of the addressing weights.

**Parallelizable recurrent networks.** Recent work has focused on designing recurrent neural networks that support parallel training while retaining efficient sequential inference, often by modeling sequential dependencies through linear recurrence relations. Early studies explored recurrent networks with linear recurrent updates (Bradbury et al., 2017; Lei et al., 2018), and concurrent work showed that such models can be parallelized using scan algorithms (Martin & Cundy, 2018). Subsequent work drew inspiration from state-space models to develop linear recurrent architectures (Gu et al., 2020; 2022a;b; Smith et al., 2023), leading to the Mamba family of models (Gu & Dao, 2024; Dao & Gu, 2024), which achieve strong efficiency through scan parallelism, linear-time inference, and competitive performance in language modeling. Architectures not directly tied to state-space formulations have also been proposed, such as the Linear Recurrent Unit (LRU) (Orvieto et al., 2023) and linear attention (Katharopoulos et al., 2020), while more recent work extends parallelization techniques to more complex recurrences, including Titans (Behrouz et al., 2026), which introduces memory updates based on gradient optimization steps, and DeltaNet (Yang et al., 2024; 2025), which uses the delta rule for associative memory updates. Other notable parallelizable recurrent architectures include RetNet (Sun et al., 2023), RWKV (Peng et al., 2023), Griffin (De et al., 2024), and xLSTM (Beck et al., 2024). Most closely related to our work, Feng et al. (2024) revisited LSTM (Hochreiter & Schmidhuber, 1997) and GRU (Cho et al., 2014), introducing simplified variants named minLSTM and minGRU that remove temporal dependencies in state updates and enable scan parallelization. Our work extends these ideas to the NTM architecture.

**Expressiveness of neural sequence models.** Several works have studied the expressive power of sequence modeling architectures, particularly those used by language models, by relating them to formal language classes and classical models of computation. It is well established that standard recurrent neural networks correspond to finite automata and thus recognize regular languages (Siegelmann, 1996; Horne & Hush, 1996; Weiss et al., 2018). More recently, research has examined linear recurrent architectures (Merrill et al., 2024; Sarrof et al., 2024), showing that specific classes of such models can exactly recognize regular languages (Grazzi et al., 2025; Siems et al., 2025). Transformers, in contrast, have been shown to have limited state-tracking capabilities, as their expressive power is restricted to constant-depth circuit classes (Merrill & Sabharwal, 2023). However, autoregressive chain-of-thought generation can overcome many of these limitations (Feng et al., 2023; Li et al., 2024b). In this setting, transformers can learn state tracking, although this requires specialized training objectives (Huang et al., 2025), with formal bounds on the number of reasoning steps (Amiri et al., 2025). Going further, autoregressive models have been shown to be capable of universal computation (Schuurmans et al., 2024), including softmax transformers (Jiang et al., 2025), although these results may depend on specific positional encodings or structured autoregressive formats not typically found in training data. Our work takes a practical perspective on these ideas: we present a parallelizable architecture inspired by a classical model of computation and evaluate its ability to generalize to unseen sequence lengths on autoregressive algorithmic tasks.

## 9 Conclusion

We introduce Parallelizable Neural Turing Machine (P-NTM), a parallelizable simplification of the original NTM whose core operations are reformulated as associative operators, enabling efficient parallel computation via scan algorithms while replacing the recurrent controller with autoregressive control through output

tokens. We additionally develop a log-space parallel algorithm that ensures numerical stability over long sequences. On synthetic algorithmic benchmarks, P-NTM successfully learns state tracking, memorization, and basic arithmetic, and generalizes to sequences longer than those seen during training, matching the performance of standard NTMs while achieving speedups of an order of magnitude on long inputs. Thus, P-NTM contributes toward the development of efficient neural architectures capable of expressing a broader class of algorithms.

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

## A   Parallel Execution and Complexity Analysis

Algorithm 3 presents the full multi-head P-NTM forward pass described in Section 3.

---

**Algorithm 3** $\boldsymbol{y}_1, \ldots, \boldsymbol{y}_T \leftarrow \texttt{MultiHeadPNTM}(\boldsymbol{x}_1, \ldots, \boldsymbol{x}_T \mid \mathcal{W})$

---

**Input:** $\boldsymbol{x}_1, \ldots, \boldsymbol{x}_T$, where $\boldsymbol{x}_t \in \mathbb{R}^d$
**Output:** $\boldsymbol{y}_1, \ldots, \boldsymbol{y}_T$, where $\boldsymbol{y}_t \in \mathbb{R}^d$
**Hyperparameters:** $H$, number of heads
**Parameters:** $\mathcal{W}$, consisting of
    $\boldsymbol{W}_r^h \in \mathbb{R}^{3 \times d}$, $\boldsymbol{W}_w^h \in \mathbb{R}^{3 \times d}$, $\boldsymbol{W}_u^h \in \mathbb{R}^{(n/H) \times d}$ for $h = 1, \ldots, H$;
    $\boldsymbol{W}_m \in \mathbb{R}^{n \times n}$ and $\boldsymbol{W}_o \in \mathbb{R}^{d \times (Hn)}$

1: **parallel for** $h \leftarrow 1$ **to** $H$ **do**
2:     **parallel for** $t \leftarrow 1$ **to** $T$ **do**                ▷ Control
3:         $\boldsymbol{s}_t^{r,h} \leftarrow \mathrm{softmax}(\boldsymbol{W}_r^h \boldsymbol{x}_t)$
4:         $\boldsymbol{s}_t^{w,h} \leftarrow \mathrm{softmax}(\boldsymbol{W}_w^h \boldsymbol{x}_t)$
5:         $\boldsymbol{u}_t^h \leftarrow \boldsymbol{W}_u^h \boldsymbol{x}_t$
6:     **end parallel for**
7:     $\boldsymbol{a}_0^{r,h}, \ldots, \boldsymbol{a}_{T-1}^{r,h} \leftarrow \texttt{ConvShift}(\boldsymbol{s}_1^{r,h}, \ldots, \boldsymbol{s}_{T-1}^{r,h})$          ▷ Addressing
8:     $\boldsymbol{a}_0^{w,h}, \ldots, \boldsymbol{a}_{T-1}^{w,h} \leftarrow \texttt{ConvShift}(\boldsymbol{s}_1^{w,h}, \ldots, \boldsymbol{s}_{T-1}^{w,h})$
9:     $\boldsymbol{M}_1^h, \ldots, \boldsymbol{M}_T^h \leftarrow \texttt{MemoryWrite}(\boldsymbol{a}_0^{w,h}, \ldots, \boldsymbol{a}_{T-1}^{w,h}, \boldsymbol{u}_1^h, \ldots, \boldsymbol{u}_T^h)$      ▷ Writing
10: **end parallel for**
11: **parallel for** $t \leftarrow 1$ **to** $T$ **do**
12:     $\tilde{\boldsymbol{M}}_t \leftarrow \boldsymbol{W}_m \left[ \boldsymbol{M}_t^{1\top} \ldots \boldsymbol{M}_t^{H\top} \right]^\top$            ▷ Mixing
13:     $\boldsymbol{r}_t \leftarrow \left[ \tilde{\boldsymbol{M}}_t^\top \boldsymbol{a}_{t-1}^{r,1} \quad \cdots \quad \tilde{\boldsymbol{M}}_t^\top \boldsymbol{a}_{t-1}^{r,H} \right]$           ▷ Reading
14:     $\boldsymbol{y}_t \leftarrow \boldsymbol{W}_o \boldsymbol{r}_t$                  ▷ Output
15: **end parallel for**
16: **return** $\boldsymbol{y}_1, \ldots, \boldsymbol{y}_T$

---

Table 3 summarizes the work and parallel depth of each stage, and the total space complexity, of P-NTM execution compared with the standard NTM. We derive each entry below.

**Control.** Each of the $H$ head pairs computes two shift vectors via $3 \times d$ projections and one update vector via an $(n/H) \times d$ projection, giving $O(d \cdot (H + n))$ work per timestep. All positions are independent, so the depth is $O(1)$.

**Addressing.** Each head's shift vectors are converted into addressing weights via FFT-based circular convolution in log space (Section 3.2). For each of the $2H$ heads, the $T$ shift kernels are padded to length $m$ and transformed via FFT at $O(m \log m)$ cost each, accumulated via prefix sum at $O(\log T)$ depth, and inverse-transformed, yielding $O(T \cdot H \cdot m \log m)$ total work.

**Memory writes.** Each scalar element of the memory matrix follows a first-order linear recurrence. Since each head writes $n/H$ dimensions per cell, there are $H \cdot m \cdot (n/H) = mn$ independent scalar recurrences, each computed over $T$ steps by a parallel scan: $O(T \cdot mn)$ work, $O(\log T)$ depth.

**Memory mixing.** The projection $\boldsymbol{W}_m \in \mathbb{R}^{n \times n}$ is applied independently to each of the $m$ cells at each timestep: $O(T \cdot mn^2)$ work, $O(1)$ depth.

**Memory reads and output projection.** Each read computes a weighted sum over $m$ cells of dimension $n$ per head: $O(T \cdot H \cdot mn)$ work. The output projection $\boldsymbol{W}_o \in \mathbb{R}^{d \times Hn}$ costs $O(T \cdot H \cdot nd)$. Both stages are fully parallel across positions.

**Totals.** The overall depth is $O(\log T)$, determined by the addressing and memory write scans. The total work is $O(T \cdot H \cdot (m \log m + mn + nd) + T \cdot mn^2)$, dominated by addressing, memory reads, output projection, and memory mixing. Relative to the sequential NTM's $O(T \cdot H \cdot (mn + nd))$, the P-NTM incurs a logarithmic factor $O(\log m)$ in addressing from the FFT and an additive $O(T \cdot mn^2)$ term from the mixing projection.

The standard NTM total omits the cost of its recurrent controller, as the P-NTM lacks one but typically relies on an external sequence model to produce its inputs.

**Space.** Parallel computation requires $O(T \cdot m(n + H))$ space, since all $T$ memory matrices, each $m \times n$, and addressing weight vectors, $2H$ vectors of length $m$, must be materialized simultaneously. The sequential NTM stores only the current state and uses $O(m(n + H))$ space.

Table 3: Work, parallel depth, and space for each stage of P-NTM execution compared with the standard NTM. $T$: sequence length, $H$: number of heads, $m$: memory size, $n$: cell size, $d$: input dimension.

| Stage | Work | Depth | Space |
|---|---|---|---|
| Control | $O(Td \cdot (H + n))$ | $O(1)$ | |
| Addressing (FFT conv.) | $O(T \cdot H \cdot m \log m)$ | $O(\log T)$ | |
| Memory writes (scan) | $O(T \cdot mn)$ | $O(\log T)$ | |
| Memory mixing | $O(T \cdot mn^2)$ | $O(1)$ | |
| Memory reads | $O(T \cdot H \cdot mn)$ | $O(1)$ | |
| Output projection | $O(T \cdot H \cdot nd)$ | $O(1)$ | |
| **Total (P-NTM)** | $O(T \cdot H \cdot (m \log m + mn + nd) + T \cdot mn^2)$ | $O(\log T)$ | $O(T \cdot m(n + H))$ |
| **Total (Standard NTM)** | $O(T \cdot H \cdot (mn + nd))$ | $O(T)$ | $O(m(n + H))$ |

# B    Experimental Details

## B.1    Algorithm Learning and Representation Experiments

### B.1.1    Task Definitions

**Parity Check.**    Given a binary string composed of $a$s and $b$s, the task is to determine whether the number of $a$s is odd or even. Given an input $x \in \{a, b\}^n$, the output is $y \in \{0, 1\}^n$, where the $k$th symbol $y_k$ is the parity of the count of $a$s in the prefix $x_1 \ldots x_k$. The final symbol $y_n$ therefore encodes the parity of the entire string.

**Cycle Navigation.**    The goal is to simulate movement on a 5-state cycle using operations for staying put, moving forward (i.e., increment), or moving backward (i.e., decrement). Let $\mathcal{S} = \{0, \ldots, 4\}$. The input is $x \in \{s, i, d\}^n$, where $s$ denotes "stay," $i$ denotes "increment," and $d$ denotes "decrement," all with wrap-around at the boundaries. The output is $y \in \mathcal{S}^n$, where each $y_k$ corresponds to the state reached after applying operations $x_1, \ldots, x_k$ starting from state 0.

**Reverse String.**    The goal is to reverse the input string. For $x = x_1 \ldots x_n \in \{0, 1\}^n$, the output is $y = x_n \ldots x_1$.

**Duplicate String.**    The goal is to repeat the input string twice with no separator. Given $x = x_1 \ldots x_n \in \{0, 1\}^n$, the output is $y = xx = x_1 \ldots x_n x_1 \ldots x_n$.

**Modular Arithmetic.**    The goal is to evaluate a modular arithmetic expression involving addition, subtraction, and multiplication modulo 5. Given an input expression that interleaves operators in $\{+, -, \times\}$ with integers in $\{0,1,2,3,4\}$ (e.g., $1 \times 2 + 4$),[9] the goal is to compute its value by maintaining partial products and partial results while scanning the expression from left to right in a single pass. The output takes the form $y = s_1 \ldots s_n v$, where each $s_i$ is a triple of symbols $h_1 h_2 h_3$ representing an intermediate step, and $v$ is the final result of the expression modulo 5. Here, $h_1$ encodes the sign of the current partial product, $h_2$ encodes the value of this partial product modulo 5, and $h_3$ encodes the current partial result of the expression (also

---

[9]Because expressions alternate operand and operator tokens, inputs must have odd length. If an even length is requested, we generate an input one token longer instead.

modulo 5). For example, the triple +12 indicates that the partial product is positive with value 1, and the running value of the expression at that step is 2.

**Binary Addition.** The goal is to add two binary numbers written in little-endian order and output their sum, also in little-endian form. The input is a string of the form $x = b_1 + b_2$, where $b_1 \in \{0, 1\}^m$ and $b_2 \in \{0, 1\}^n$. The output is $y = \text{bin}(\text{int}(b_1) + \text{int}(b_2))$, the binary representation of the integer sum, also in little-endian order.

### B.1.2 Architecture Descriptions

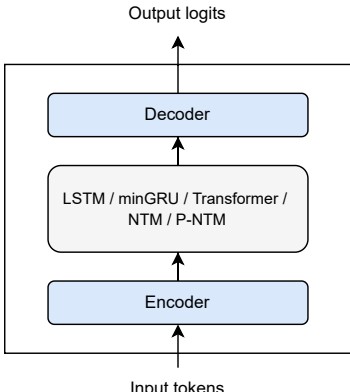

Figure 5: High-level structure shared by all evaluated models. An encoder maps input tokens to embeddings, a sequence-processing module transforms them, and a decoder produces vocabulary logits.

All models used in the experiment share the same high-level structure, illustrated in Figure 5. Each model consists of (i) an encoder that maps input tokens to embeddings, (ii) a sequence-processing module, and (iii) a decoder that produces logits over the vocabulary via linear projection. The only component that differs across models is the sequence-processing module. Detailed configurations for each architecture are provided below, and Table 4 summarizes the hyperparameters.

**LSTM.** Consists of a single LSTM layer applied to the embedded token sequence, with both the embedding dimension and the LSTM hidden size set to 192. The LSTM produces a sequence of hidden states, which are linearly projected by a decoder layer to generate output logits at each timestep.

**minGRU.** Comprises a stack of 5 identical residual minGRU blocks with hidden size 64. Each block includes (i) a pre-normalized minGRU sequence-mixing sublayer with a state-expansion factor of 2 (yielding a state size of 128), followed by (ii) a pre-normalized position-wise feedforward sublayer with 4× expansion and GELU activation. Both sublayers use residual connections (see Figure 6).

**Transformer.** Consists of 5 identical residual Transformer decoder blocks with embedding width 64. Each block contains (i) a pre-normalized multi-head causal self-attention sublayer with 8 heads, followed by (ii) a pre-normalized position-wise feedforward sublayer with 4× expansion and GELU activation; both sublayers use residual connections. Positional information is incorporated via one of the following: no explicit positional encoding (NoPE), RoPE (rotary embeddings with $\theta = 10{,}000$), ALiBi, or FIRE (learned bias network with hidden width 32, initialized with $c_0 = 0.1$, $L_0 = 512$, and $\varepsilon = 10^{-6}$). See Figure 6.

**NTM.** Consists of an NTM controller with embedding width 104 and an external memory composed of cells of size 32. At each timestep, the controller interacts with memory through 4 read heads and 4 write heads, producing a sequence of output representations. These are projected to token logits by a final decoder layer.

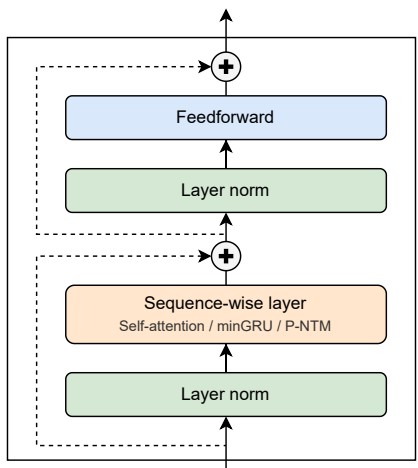

Figure 6: Residual block structure used by the minGRU, Transformer, and P-NTM models. Each block applies a sequence-wise sublayer (self-attention, minGRU, or P-NTM layer) over the full sequence, followed by a position-wise feedforward network. Both sublayers use pre-normalization and residual connections. LSTM and NTM architectures do not adopt this structure.

Table 4: Hyperparameters and parameter counts for all evaluated models. Dashes (—) indicate that a hyperparameter does not apply.

| Architecture | Hidden Size | Depth | Heads | Cell Size | Parameters |
|---|---|---|---|---|---|
| LSTM | 192 | — | — | — | ∼298 K |
| minGRU | 64 | 5 | — | — | ∼289 K |
| Transformer (NoPE) | 64 | 5 | 8 | — | ∼248 K |
| Transformer (ALiBi) | 64 | 5 | 8 | — | ∼248 K |
| Transformer (RoPE) | 64 | 5 | 8 | — | ∼248 K |
| Transformer (FIRE) | 64 | 5 | 8 | — | ∼249 K |
| NTM | 104 | — | 4 | 32 | ∼224 K |
| P-NTM | 104 | 2 | 4 | 32 | ∼260 K |

**P-NTM.** Consists of two residual blocks with embedding width 104. The first block is a minGRU block as described above. The second block is a P-NTM block, whose sequence-mixing sublayer comprises a P-NTM layer with memory cell size 32 and 4 heads, followed by a position-wise feedforward sublayer with 4× expansion and GELU activation. All sublayers are pre-normalized and use residual connections (see Figure 6).

### B.1.3 Training and Evaluation

During training, both NTM and P-NTM used a memory size of $m_{\text{train}} = 2 \times \ell_{\text{max}} + 16 = 96$, where $\ell_{\text{max}}$ is the maximum input length observed during training. This avoids positional interference from circular wraparound of the write heads. During evaluation, the memory size was increased to 256 for both models, again allocating twice the maximum input length with additional slack. For P-NTM, a stability threshold of $\tau = 0.01$ was used.

All experiments used 10 random seeds produced by NumPy's random number generator from an initial seed of 0.

### B.2 Computational Efficiency Experiments

**Model size.** For the computational efficiency experiments, the standard NTM model contains 168,140 trainable parameters. The P-NTM model contains 152,576 parameters in total, of which 147,456 correspond to the minGRU layer.

**Hardware configuration.** All experiments ran on a single NVIDIA A100 SXM4 GPU (80 GB HBM2e, 1522.0 GB/s memory bandwidth, CUDA 12.x) hosted on a system with an AMD EPYC 7532 CPU (64 logical cores) and 129 GB of RAM.

## C  Supplementary Results

### C.1  Transformer Positional Encodings

Table 5: Maximum exact match accuracy scores across runs for each task and Transformer positional encoding scheme. Best results boldfaced.

| Pos. Encoding | PC | CN | RS | DS | MA | BA |
|---|---|---|---|---|---|---|
| NoPE | 0.16 | 0.19 | 0.18 | 0.24 | 0.19 | **0.22** |
| RoPE | 0.06 | 0.07 | 0.05 | 0.05 | 0.04 | 0.06 |
| ALiBi | **0.27** | **0.56** | **0.21** | **0.65** | 0.38 | 0.20 |
| FIRE | 0.23 | 0.44 | 0.19 | 0.28 | **0.42** | 0.19 |

In Table 5, ALiBi achieves the highest accuracy on four of six tasks, with FIRE close behind and ranking first on modular arithmetic. RoPE generalizes poorly beyond the training length, with accuracy consistently below 0.07. NoPE achieves the best result on binary addition but otherwise trails ALiBi and FIRE.

### C.2  Mean and Standard Deviation Scores

Table 6: Mean ($\pm$ standard deviation) exact match accuracy across runs for each architecture and task on problems of unseen length ($\ell = 41$ to $\ell = 120$). For Transformers, the positional encoding variant with the highest maximum exact match accuracy per task is selected (see Table 5).

| Architecture | PC | CN | RS | DS | MA | BA |
|---|---|---|---|---|---|---|
| LSTM | $0.076 \pm 0.007$ | $0.030 \pm 0.011$ | $0.123 \pm 0.011$ | $0.072 \pm 0.020$ | $0.044 \pm 0.007$ | $0.092 \pm 0.015$ |
| minGRU | $0.046 \pm 0.007$ | $0.028 \pm 0.009$ | $0.075 \pm 0.027$ | $0.078 \pm 0.009$ | $0.067 \pm 0.007$ | $0.081 \pm 0.020$ |
| Transformer (best) | $0.199 \pm 0.046$ | $0.382 \pm 0.082$ | $0.161 \pm 0.025$ | $0.491 \pm 0.111$ | $0.256 \pm 0.080$ | $0.179 \pm 0.016$ |
| NTM | $0.962 \pm 0.071$ | $0.928 \pm 0.216$ | $0.982 \pm 0.023$ | $0.987 \pm 0.022$ | $0.993 \pm 0.015$ | $0.946 \pm 0.145$ |
| P-NTM | $1.000 \pm 0.000$ | $1.000 \pm 0.000$ | $0.899 \pm 0.189$ | $0.805 \pm 0.164$ | $0.988 \pm 0.029$ | $0.916 \pm 0.153$ |

Although both NTM and P-NTM achieve perfect maximum scores on all tasks (Table 2), Table 6 reveals differences in reliability. P-NTM achieves perfect mean accuracy on parity check and cycle navigation, but shows substantially lower means and higher variance than NTM on reverse string, duplicate string, and binary addition, consistent with its sharper failure mode (Section 5). NTM shows its greatest variability on cycle navigation ($0.928 \pm 0.216$), where P-NTM is perfectly consistent.

### C.3  Accuracy by Input Length

Both architectures maintain near-perfect exact match accuracy within and just beyond the maximum training length (dashed line in Figure 7), and a substantial fraction of runs sustain perfect accuracy across the full evaluated range. When performance does degrade beyond length 40, the failure modes differ: P-NTM exhibits sharp drops, with several runs collapsing to zero accuracy between lengths 70 and 120, whereas NTM degrades more gradually, typically retaining partial accuracy even at the longest evaluated lengths.

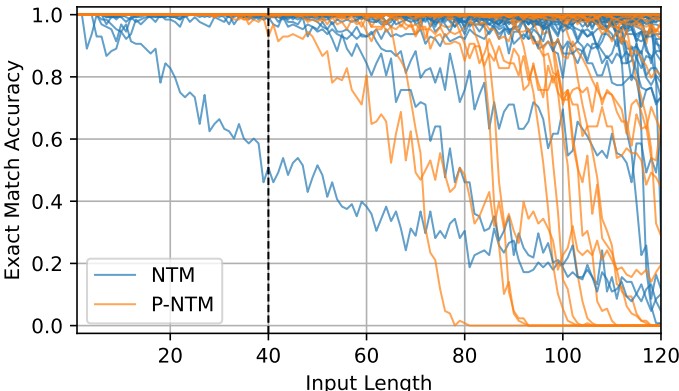

Figure 7: Exact match accuracy across varying input lengths for NTM and P-NTM. Each curve represents a separate run on a specific task. The vertical dashed line marks the maximum training length ($\ell = 40$). While NTM generally maintains non-zero accuracy at extended lengths, P-NTM exhibits sharp performance degradation, frequently dropping to zero accuracy for longer inputs.

## C.4 Training Convergence

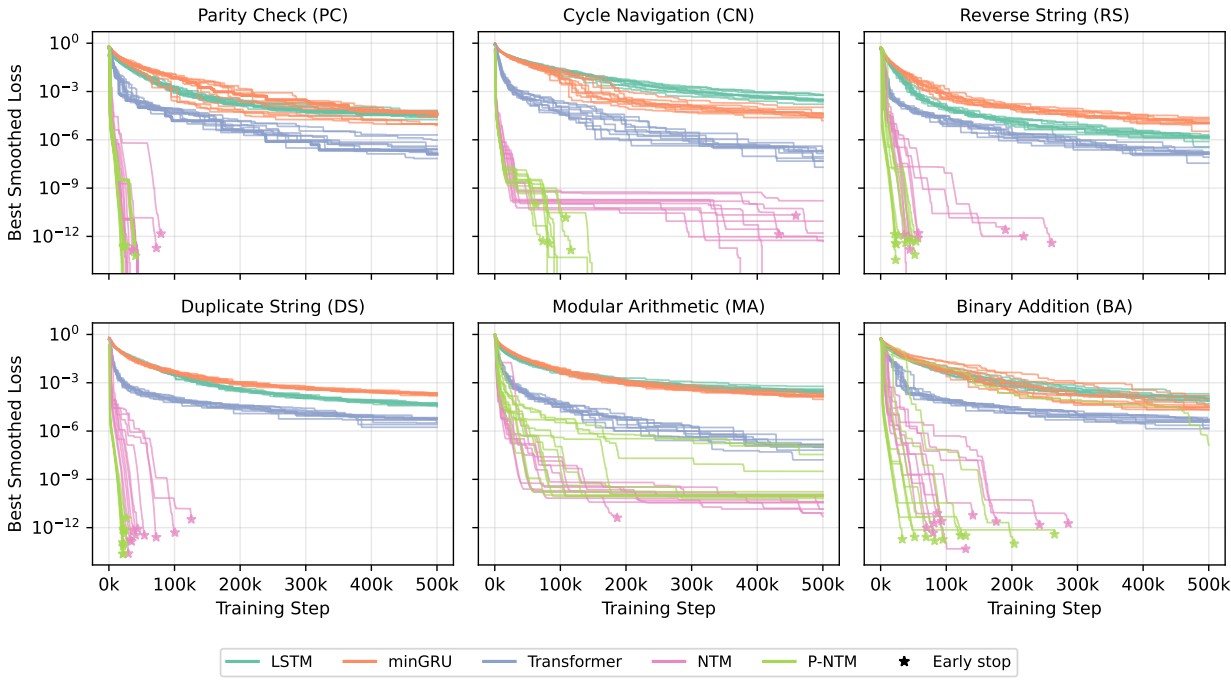

Figure 8: Training convergence across architectures and tasks. Each curve shows the running minimum of the training loss after smoothing with a rolling mean (window size 1,000 steps). Stars ($\star$) mark runs that triggered early stopping. For Transformers, the positional encoding variant with the highest maximum exact match accuracy per task is shown (see Table 5). Memory-augmented architectures (NTM and P-NTM) reach substantially lower loss floors than baselines, with P-NTM generally converging fastest.

Figure 8 plots the running minimum of the smoothed loss for each of the ten runs per architecture. Memory-augmented models (NTM and P-NTM) converge to loss values several orders of magnitude lower than the

baselines, which typically plateau around $10^{-1}$ to $10^{-6}$ despite training for the full 500,000 steps. NTM and P-NTM frequently trigger early stopping, with P-NTM converging fastest on most tasks. The gap in training loss foreshadows the generalization gap in Section 5: architectures that fail to fully minimize training loss on in-distribution sequences also fail to generalize to longer ones.

## C.5 Execution Times

Table 7: Mean execution time ($\pm$ standard deviation), measured in seconds, across batches of input sequences of varying lengths for NTM and P-NTM. The P-NTM results include both sequential and parallel execution.

| Input Length | NTM | P-NTM (Sequential) | P-NTM (Parallel) |
|---|---|---|---|
| $2^3 = 8$ | $0.0011 \pm 0.0000$ | $0.0007 \pm 0.0000$ | $0.0003 \pm 0.0000$ |
| $2^4 = 16$ | $0.0019 \pm 0.0001$ | $0.0012 \pm 0.0000$ | $0.0004 \pm 0.0000$ |
| $2^5 = 32$ | $0.0035 \pm 0.0000$ | $0.0023 \pm 0.0000$ | $0.0005 \pm 0.0000$ |
| $2^6 = 64$ | $0.0068 \pm 0.0001$ | $0.0043 \pm 0.0000$ | $0.0006 \pm 0.0000$ |
| $2^7 = 128$ | $0.0128 \pm 0.0000$ | $0.0083 \pm 0.0000$ | $0.0011 \pm 0.0001$ |
| $2^8 = 256$ | $0.0209 \pm 0.0003$ | $0.0164 \pm 0.0000$ | $0.0017 \pm 0.0002$ |
| $2^9 = 512$ | $0.0411 \pm 0.0001$ | $0.0325 \pm 0.0001$ | $0.0029 \pm 0.0000$ |
| $2^{10} = 1,024$ | $0.0818 \pm 0.0001$ | $0.0651 \pm 0.0008$ | $0.0057 \pm 0.0001$ |
| $2^{11} = 2,048$ | $0.1620 \pm 0.0001$ | $0.1305 \pm 0.0020$ | $0.0109 \pm 0.0001$ |
| $2^{12} = 4,096$ | $0.3268 \pm 0.0001$ | $0.2588 \pm 0.0007$ | $0.0185 \pm 0.0000$ |
| $2^{13} = 8,192$ | $0.6546 \pm 0.0004$ | $0.5197 \pm 0.0014$ | $0.0363 \pm 0.0000$ |
| $2^{14} = 16,384$ | $1.3079 \pm 0.0001$ | $1.0396 \pm 0.0017$ | $0.0718 \pm 0.0001$ |
| $2^{15} = 32,768$ | $2.6242 \pm 0.0019$ | $2.0857 \pm 0.0030$ | $0.1414 \pm 0.0001$ |
| $2^{16} = 65,536$ | $5.2328 \pm 0.0018$ | $4.0207 \pm 0.0200$ | $0.2832 \pm 0.0001$ |

The standard NTM and sequential P-NTM exhibit similar scaling in Table 7, with sequential P-NTM providing a consistent but modest latency reduction (approximately 1.2 seconds at the largest input size). Parallel P-NTM scales markedly differently: it processes a sequence of length $2^{16} = 65{,}536$ in 0.28 seconds, whereas the standard NTM requires 0.33 seconds for $2^{12} = 4{,}096$ elements. Within a comparable time budget, the parallel architecture thus accommodates a sixteen-fold increase in sequence length.

## C.6 Memory Access Patterns

Figure 9 shows the memory access patterns of a trained P-NTM on a parity check instance. Write Head 4 advances monotonically through memory, storing distinct symbol representations in consecutive locations with write vectors that shift in sync with the input. During output generation, Read Head 4 retrieves these locations in the original order, with read vectors reflecting the stored input rather than the current output token. This suggests that the model solves the parity task by buffering the input into memory and sequentially re-reading it to produce the output.

Figure 10 shows an analogous visualization for the reverse string task, using a run that achieved 100% exact match accuracy on unseen lengths. The reverse string model follows a similar strategy with a key difference in read direction. Write Heads 3 and 4 advance monotonically through consecutive addresses during the input phase, buffering each token into a distinct location. Read Head 2 is most informative: during input, its addressing weights advance from address 0 to address 5, mirroring the write pattern; during output generation, they reverse direction from address 5 back to address 0, producing the characteristic tent-shaped trajectory visible in the addressing heatmap. This suggests that the model retrieves stored tokens in reverse order to implement string reversal.

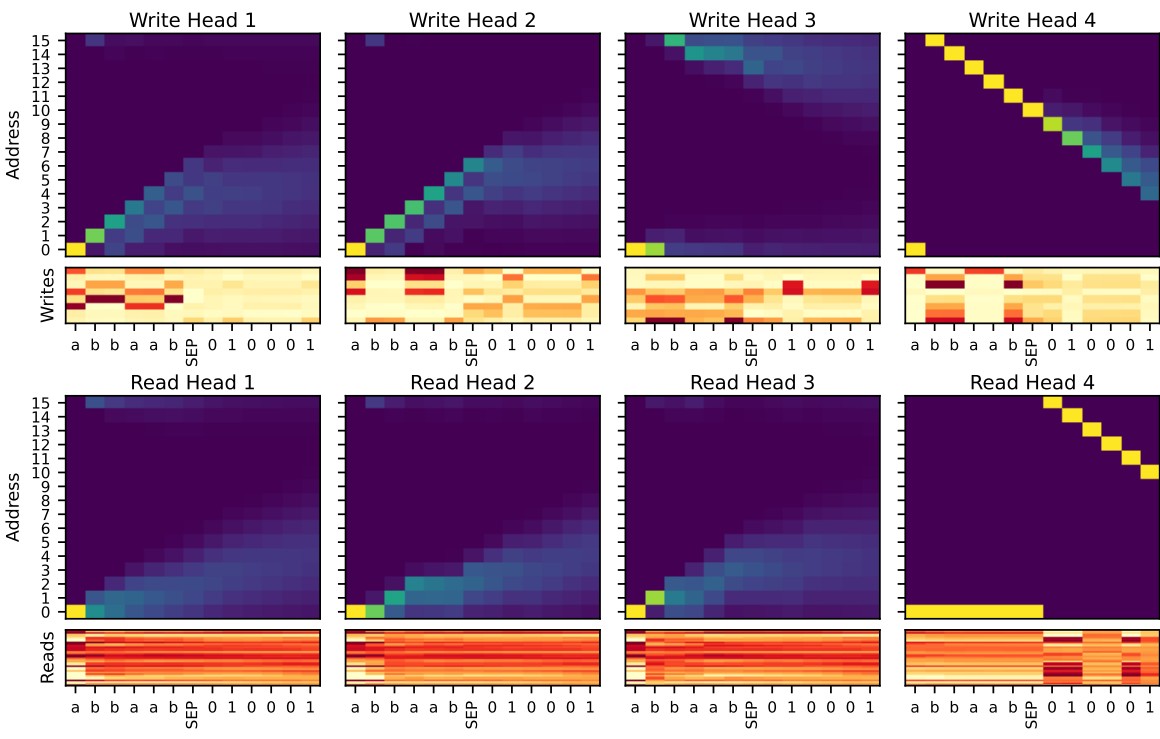

Figure 9: Memory interaction patterns of the heads of a trained P-NTM on a parity check sequence (input `abbaab`, output `010001`). The top row displays **write heads**, and the bottom row displays **read heads**. For each head, the upper panel visualizes addressing weights across memory locations over time, while the lower strip shows the read or write vector at each timestep (normalized per head). Write Head 4 and Read Head 4 exhibit the sharpest addressing weights.

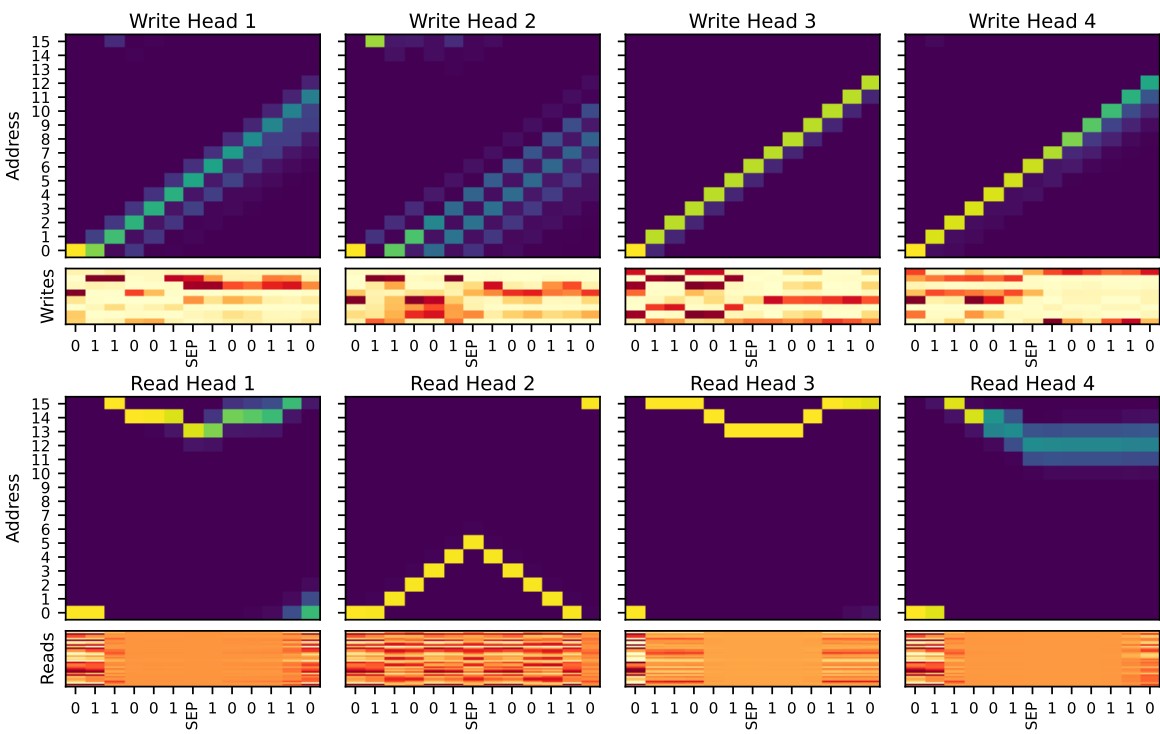

Figure 10: Memory interaction patterns of the heads of a trained P-NTM on a reverse string instance (input `011001`, output `100110`). The layout follows Figure 9. Write Heads 3 and 4 exhibit the sharpest addressing weights; Read Head 2 shows a characteristic tent-shaped trajectory, advancing through addresses during input and reversing direction during output.

