# OpenReview forum: "Parallelizable Neural Turing Machines"
_TMLR — Withdrawn by Authors_

### Review · Reviewer_Kxb6 · 2026-06-03

**Summary Of Contributions:**

The paper parallelizes the NTM by restricting to location-based addressing (reducible to FFT-based cumulative convolution) and removing the recurrent controller, with computational state tracked implicitly through autoregressive output tokens. On six synthetic tasks, P-NTM matches the standard NTM's peak generalization accuracy while achieving up to 18× training speedup. The contribution is incremental but cleanly executed.

**Audience:**

Yes

**Audience Explanation:**

Parallelizing memory-augmented networks is a natural direction that hasn't received much attention, and the finding that removing content-based addressing doesn't hurt generalization on these tasks is worth knowing.

**Claims And Evidence:**

Yes

**Claims Explanation:**

The efficiency and generalization claims are well-supported within the experimental scope. My one concern: Table 2 reports only maximum accuracy across 10 seeds, but Table 6 in the appendix shows P-NTM's mean accuracy on RS/DS/BA is 0.899/0.805/0.916 — notably lower than NTM's, with higher variance. Leading with the max gives a more favorable picture than the full results warrant.

**Requested Changes:**

The six tasks are all solvable with sequential memory access, which may explain why removing content-based addressing has no cost here. It's unclear whether this holds for tasks requiring associative lookup. Some discussion of this scope limitation would help readers calibrate the contribution.
Table 2 reports only maximum accuracy across 10 seeds. Table 6 in the appendix reveals P-NTM's mean accuracy on RS/DS/BA is 0.899/0.805/0.916 with substantially higher variance than NTM. This should be in the main results, not buried in the appendix.
The evaluated P-NTM always includes a minGRU prefix layer. Without ablating this component, it's hard to know how much of the result actually comes from P-NTM itself.
Section 3.4 and the abstract claim computational equivalence between autoregressive control and the recurrent controller. The authors themselves admit in Section 7 that the formal model is unknown — the abstract should reflect this.

---

### Review · Reviewer_CGxW · 2026-06-23

**Summary Of Contributions:**

The paper proposes P-NTM, a simplification of the Neural Turing Machine that removes content-based addressing and the recurrent controller state, retaining only location-based addressing with controls computed solely from the current input. This restructuring makes the core recurrences associative, enabling parallel-scan execution. The authors also contribute a numerically stable, log-space FFT-based parallel algorithm for the shift-addressing recurrence and a parallel scan for the memory-write recurrence. Experiments on six synthetic algorithmic tasks show P-NTM matches a revisited, numerically stabilized NTM in peak length-generalization accuracy, while parallel execution yields up to ~18x speedup over the standard NTM at long sequence lengths.

**Strengths**
1. The log-space stabilization for the FFT-based shift convolution is solid engineering and solves a real underflow problem that would otherwise break long-sequence training.
2. The fix to the original NTM's sharpening instability via log-sum-exp is a useful contribution on its own, independent of the main P-NTM proposal, and improves the baseline used for comparison.

**Weaknesses**
1.  The headline P-NTM results are obtained using a minGRU layer prepended to P-NTM, not pure P-NTM as defined in Section 3. In absence of an ablation with bare P-NTM, it cannot be ascertained how much of the reported generalization is attributable to P-NTM's own mechanism versus the recurrent minGRU module.
2. No empirical comparison is provided against Baby-NTM, even though the authors themselves note it to be the closest prior simplification of NTM. Similarly, DeltaNet-family architectures are discussed at length but not benchmarked, despite being most relevant for the state-tracking claims made.
3. The efficiency experiments are conducted with single read-write head, whereas accuracy experiments use 4 heads. Since FFT-based addressing cost scales with the number of heads, the reported speedup figures may not generalize to the multi-head setting actually used elsewhere in the paper.
4. Space complexity is flagged as the main practical tradeoff of parallel execution, but no actual memory/VRAM measurements are reported to substantiate this claim.

**Audience:**

Yes

**Audience Explanation:**

Yes, the parallelisation approach and the log-space numerical stability tricks would be of interest to the segment of TMLR's audience working on parallelisable recurrent architectures and memory-augmented networks, since these contributions are reusable independent of the specific P-NTM proposal.

**Claims And Evidence:**

No

**Claims Explanation:**

My biggest concern is that the headline P-NTM model used in all the accuracy experiments is not really "P-NTM" as defined in Section 3. The architecture as defined there has control depending solely on the current input token, with no recurrence at all in the control mechanism. But the actual model being tested prepends a minGRU layer in front of P-NTM, and the authors say plainly that this is because otherwise the control "would otherwise only consider the current token." This is an important admission buried in the experimental setup, and it means we genuinely do not know how much of the perfect generalization being reported comes from P-NTM's own associative mechanism versus the recurrent minGRU sitting in front of it. Without an ablation showing bare P-NTM's numbers, I cannot agree with the claim that "the proposed architecture matches the original in generalization." It may be that minGRU-plus-P-NTM matches it, which is a different and weaker claim.

Second, the related work section is quite thorough. Baby-NTM, D-NTM, DNC, Neural GPU, nnTM are all discussed, and Baby-NTM in particular is described by the authors themselves as closest to their work since it also drops content-based addressing. But none of these are run as empirical baselines. Given that Baby-NTM already explores the location-only addressing simplification, I think a direct comparison is needed to know what the parallelization contribution is adding on top, separate from the addressing simplification itself. Right now the only baselines are LSTM, minGRU, Transformer variants, and a revisited NTM. None of which test the specific design space this paper is contributing to.

Third, I am not fully convinced by the efficiency benchmark either. The speedup numbers are measured with a single read-write head pair, but the accuracy experiments use 4 heads. Since the FFT-based addressing cost scales with the number of heads, I would like to see the speedup curve for the same multi-head configuration that is actually used to get the accuracy results. As it stands, the two experiments are testing different configurations, and I worry the 18x number may not hold once you are at H=4.

**Requested Changes:**

See weaknesses

---

### Review · Reviewer_ASEs · 2026-06-26

**Summary Of Contributions:**

This paper introduces Parallelizable NTM method by reformulating the NTM operations by associative property of many multiplcations and binary tree based data storing scheme. Empirical evaluations show that P-NTM performs on par with NTMs while P-NTM speeds up 18 times than its original version (the speed up is proportional to the exponent of input size). So, the moderate input increase (x2 or x10) may not enjoy the huge computational gain.

**Additional Comments:**

N/A

**Audience:**

Yes

**Audience Explanation:**

If someone wants to use NTM with computational efficiency (especially if the input size is huge), this method should be very useful.

**Broader Impact Concerns:**

There are no immediate ethical concerns or direct potential for malicious deployment.

**Claims And Evidence:**

Yes

**Claims Explanation:**

They evaluate the P-NTM on 6 different tasks, matching the accuracy to the original NTM with computational efficiency.

**Requested Changes:**

1. Show the performance gain alongside with computational complexity (maybe in wall clock time or FLOPs), not showing them in separate plots.
2. Explain the tree based scheme in figure.
3. Contrast the architectural difference to NTM in Fig 2
4. Quantitative evaluation on sparse/missing intermediate steps of reasoning
5. Sensitivity analysis on $\tau$.

---

### Note · Authors · 2026-07-09

**Comment:**

We thank the reviewers for their insightful feedback. Their comments were valuable and gave us a clear sense of how to improve our work.

After careful consideration, we found that the changes needed were too substantial to make within the review window. We believe the paper would benefit significantly from additional time to develop these ideas properly, and for this reason we are withdrawing the submission.

We will incorporate the feedback into an updated submission with more complete experiments, results, and discussion. We will be paying special attention to the points related to experiments in the absence of intermediate steps, the inclusion of more relevant baselines, ablations over the number of heads (for both generalization and efficiency), and clarifying that our claims concern P-NTM as a backbone layer rather than in isolation.

Once again, we thank the reviewers for their time and thoughtful engagement.

**Withdrawal Confirmation:**

I have read and agree with the venue's withdrawal policy on behalf of myself and my co-authors.